# Investigating the Impact of Fly Ash on the Strength and Micro-Structure of Concrete during Steam Curing and Subsequent Stages

**DOI:** 10.3390/ma16041326

**Published:** 2023-02-04

**Authors:** Yun Duan, Qicai Wang, Zhaofei Long, Xiaoping Wang

**Affiliations:** 1Civil Engineering Department, Lanzhou Jiaotong University, Lanzhou 730070, China; 2National and Provincial Joint Engineering Laboratory of Road & Bridge Disaster Prevention and Control, Lanzhou Jiaotong University, Lanzhou 730070, China

**Keywords:** fly ash, compressive strength, microstructure, steam curing, manufactured sand concrete, pozzolanic reaction

## Abstract

Steam curing, a common way of curing precast concrete, can greatly improve its 1-day strength. However, the thermal effect of prolonged high-temperature curing can negatively impact the concrete’s performance, thus compromising production of precast products in cold regions. Fly ash (FA) is used as a supplementary cementitious material to improve part of the properties of concrete. In this paper, we investigated the effect of FA (10~30%) on the compressive strength and microstructure of manufactured sand concrete at the steam curing and later stages. Specifically, we analyzed the behavior of FA in the constant temperature phase under steam curing. Results indicated that the pozzolanic reaction of FA started to occur at 24 h of constant temperature curing. Early hydration under steam curing produces a large amount of Ca(OH)_2_, causing the pozzolanic reaction of FA to occur significantly earlier, and the high pH value of the solution and the fibrous mesh structure of the FA surface promote the pozzolanic reaction. The addition of 30% FA to manufactured sand concrete causes a significant reduction in early strength under steam curing, which is not beneficial to the formwork removal and tensioning of precast members. Notably, manufactured sand concrete with 20% FA under steam curing had the highest late strength. The filling effect of FA and the additional gel produced by the pozzolanic reaction would result in the reduction in large pore content, refinement of pore size, improvement of microstructural compactness, and increase in gel system strength. Therefore, the addition of 20% FA to the manufactured sand concrete can improve the long-term strength, which is beneficial to the production of precast beams in cold regions.

## 1. Introduction

Precast concrete products play an important role in infrastructural development. The sleepers and precast beams used in China‘s high-speed railway are precast prestressed concrete components [1,2]. To improve the productivity of precast products, steam curing is often used to shorten the time required to reach demoulding and prestress tensioning strength [3,4,5]. Typically, the temperature of steam curing of concrete is between 50 °C and 70 °C [6,7], but higher than 70 °C after curing. Studies have shown that delayed ettringite formation (DEF) not only causes volume expansion, but also leads to concrete cracking [8,9,10]. Therefore, increasing curing temperature rarely improves the early strength of precast girder plants. The duration of the constant temperature phase in steam curing is usually controlled at 4–12 h, and the strength at the end of steam curing can generally meet the demolding strength (30 MPa) [1,4,11]. However, it is difficult to achieve the requirements for prestress tensioning strength (50 MPa) [4,12]. Although elongating the duration of constant temperature under steam curing can make the strength at the end of steam curing meet the prestress tensioning requirements, it has also been shown to cause an uneven distribution of hydration products and structural looseness, and increase porosity and large pore content [13,14,15,16]. Consequently, these phenomena have been shown to reduce concrete’s long-term strength, thereby seriously affecting the performance and service life of concrete structures [6,11,17,18]. Winter in the north of China is cold and lasts a long time. For precast components produced under this environment, engineers expect the strength of the concrete at the end of steam curing to meet prestress tensioning requirements. This is to both increase the productivity of the precast plant and reduce production costs and energy consumption by eliminating subsequent curing.

Reducing the harmful effects of prolonged steam curing on concrete properties and microstructure is imperative to achieving the engineers’ goals. Fly ash (FA), a solid waste from coal burning, is widely used in concrete as a typical supplementary cementitious material [19,20]. Partial replacement of cement by FA not only makes wise use of solid waste resources, but also reduces carbon dioxide emissions and energy consumption [21]. Externally, FA particles appear spherical, a phenomenon that facilitates dispersion of cement particles and promotes the sliding of neighboring particles, while the ball-bearing effect enhances the fluidity of fresh concrete [22,23,24]. FA has a particle size that ranges between 1–100 μm. FA‘s microspheres provide nucleation sites for C-S-H gels at the early stage of hydration, which promotes the hydration of the cement [25,26]. The inert FA acts as a filler to reduce the large pore content [19,27]. Glassy FA contains reactive SiO_2_ and Al_2_O_3_, which can react with calcium hydroxide (CH) produced by cement hydration to produce more calcium silicate hydrate (C-S-H), calcium aluminate hydrate (C-A-H) and calcium aluminum silicate hydrate (C-A-S-H). These gels have been shown to improve the late strength and reduce the capillary content of concrete [28,29,30]. Analysis of FA‘s positive effects indicates that they have the potential to reduce the negative effects of the thermal effects of steam curing. Therefore, it is necessary to evaluate the behavior of FA under steam curing.

FA‘s low reactivity, coupled with its delayed effect on setting time [20,26], makes it less than 35% of mass of cement replacement in concrete [19,31]. Using steam curing for FA concrete not only improves FA‘s reactivity, increases the concrete‘s early strength [1,32,33,34] and completes the demoulding of precast products as early as possible, but also increases the amount of FA used [31]. Yazıcı et al. [35] studied the compressive strength of concrete with different FA (class C) contents after 5 h of constant temperature (65 °C) under steam curing, and found that the 1-day strength decreased with the increase in FA content. However, after the substitution of FA exceeded 30%, the concrete had a strength less than 30 MPa for 1 day, which was not conducive to the removal of precast formwork. The strength first increased at 28 days, then decreased with increasing FA content. During this process, concrete with 20% FA had the greatest strength. However, the authors did not explain this phenomenon. Jiang et al. [36] showed that the late strength of concrete with 30% FA (class F) was lower than that of pure cement at a constant temperature (60 °C) of 8 h under steam curing. With the FA content increased, the late strength was reduced. Previous studies have shown that high-temperature curing accelerates the pozzolanic reaction of FA, which causes a release of Al^3+^ and Si^4+^ into the C-S-H gel structure and connected silicon short chains, thereby increasing the content of silicate tetrahedra coordinated with aluminate tetrahedra, and increasing in the polymerization of C-S-H gels [37]. Chindaprasirt et al. [38] found that the 28d compressive strength of FA (class C) concrete was higher than that of cement concrete.

As mentioned above, although much research has been conducted in this field, several limitations still exist. For example, controlling constant temperature duration in the current steam curing regime does not effectively meet the tensioning requirements of precast beams, especially for the production of precast beams in cold regions. Studies on FA concrete have mainly focused on normal temperature curing, and there is a lack of research on the strength and microstructure of FA concrete under steam curing, especially the steam curing process, because it is a critical period for the formation of concrete properties and microstructure. It is certain that high temperature curing can have harmful effects on the long-term strength and microstructure of concrete, but only a few studies have explored how to reduce the thermal damage effect.

In this paper, we targeted production of precast beams in cold regions with the aim of understanding the compressive strength and microstructure of FA concrete at the constant temperature stage of steam curing. Next, we discuss the effect of FA on the long-term strength of steam-cured manufactured sand concrete. We used X-ray diffraction (XRD) analysis, scanning electron microscopy (SEM) and nuclear magnetic resonance (NMR) techniques to investigate the properties, microstructure and pore structure of the hydration products and FA’s pozzolanic reaction products under steam curing.

## 2. Materials and Methods

### 2.1. Materials

The Portland cement (P Ⅱ 52.5) produced in Lanzhou Qilianshan Cement Plant, with a specific surface of 351 m2/kg, meets the requirements of GB 175-2007 [39]. Class F type of FA was generated by burning bituminous coal, which was obtained from Xigu Power Plant, China. The chemical composition of cement and fly ash was shown in Table 1. The external characteristics of FA particles were observed via SEM, while their chemical composition was analyzed by energy dispersive X-ray spectroscopy (EDS), as shown in Figure 1. FA‘s phase composition was measured by XRD analysis (Figure 2), whereas particle size distributions of cement and FA were measured by laser diffraction granulometry, respectively. Results were shown in Figure 3. FA‘s D50 value was smaller than that of cement. The coarse aggregate was 5–20 mm particle-sized crushed stone with 0.4% powder content and 12.4% crushing index. The fine aggregate was limestone manufactured sand with a fineness modulus of 2.91, a powder content of 5.6% and a methylene blue value of 0.8%. All these met the requirements of GB/T 14684 [40]. The polycarboxylate superplasticizer with solids content of 23.6% and water reduction rate of 27% was used to improve the workability of the manufactured sand concrete and achieve the expected slump requirements [41].

### 2.2. Concrete Mixing and Specimen Preparation

Normally, a maximum of 35% by mass of FA can be used to replace cement in concrete [19,31], so the admixture of FA by mass in the concrete mixes of this study was 10%, 20% and 30%. Details of the mixing proportions of the concrete with different FA contents are shown in Table 2. This mix proportion (F0) was the common ratio for the precast beams of the high-speed railway in China [41].

All concrete specimens were prepared and tested at room temperature (20 ± 2 °C). Briefly, materials were first weighed, according to the proportions of the mixture, then the crushed stone, manufactured sand, cement and fly ash were poured into the forced mixer in turn and mixed for 90s. Next, water and polycarboxylate superplasticizer were added into the mixture and the mixing continued for 2 min. The slump of fresh concrete was 160–190 mm, which met the production requirements of precast beams in the plant. The fresh concrete was loaded into a cube mold, measuring 100 mm × 100 mm × 100 mm, for compressive strength and NMR testing. The concrete specimens loaded into the molds were placed on a vibrating table and compacted to reduce air bubbles. The specimens were then subjected to both steam curing and standard curing. SEM specimens were prepared by selecting the mortar portion of fresh concrete to reduce the negative effects of the concrete crushing sampling process on the microstructure of the cement matrix. It uses the same curing regime as concrete.

### 2.3. Curing Regimes

The steam curing regime commonly used for the production of precast beams for the high-speed railway in China [12] was used in this study. Curing was completed at a constant temperature of 50 °C, with other control parameters shown in Figure 4. To investigate the effect of FA in the constant temperature stage of steam curing, the experiment was controlled by the constant temperature curing time, using the A steam curing regime (Figure 4). The test time was 6 h, 12 h, 18 h, 24 h, 30 h and 36 h of constant temperature. To analyze the effect of FA on long-term performance and microstructure of steam-cured concrete, the B steam curing regime (Figure 4) was used. The steam curing equipment was shown in Figure 5. After the end of steam curing, the concrete was continued to be cured under standard curing conditions until the test age. Standard curing was used as a reference group, with testing completed for 28 and 90 days. The standard curing temperature was 20 ± 2 °C and relative humidity was 96% or more.

### 2.4. Methods

#### 2.4.1. Compressive Strength

The compressive strength of concrete was tested according to the China standard GB/T 50081 [42] at a loading rate of 0.6 MPa/s. Three specimens were tested under each age of every mixture, then an average value of the compressive strength was calculated.

#### 2.4.2. Scanning Electron Microscopy

The concrete microstructure was observed by Scanning electron microscopy (SEM), with a Zeiss Gemini SEM 500. Briefly, test specimens were obtained from mortar specimens using a small cutting machine, then a plane surface without any grinding treatment was chosen as the test area. To obtain clear microstructure photos, the specimens were sprayed with gold to increase their electrical conductivity.

#### 2.4.3. X-ray Diffraction

X-ray diffraction (XRD) method was used to analyze the phase composition of the specimen. The specimen was taken from the slurry portion of the crushed concrete and ground into a fine powder The test equipment was a Rigaku Miniflex 600 with an X-ray source of 60 V. The scan speed was 2°/min and the 2-theta range was 5° to 65°.

#### 2.4.4. Pore Structure

Nuclear magnetic resonance techniques have been widely used to test the pore water content, pore structure and pore size distribution of cement and concrete materials [43,44]. Based on quantum mechanical property, hydrogen nuclei (^1^H) always spin around their own axis at a certain frequency, which is equivalent to a small constant magnetic field. In the presence of an external fixed main magnetic field, part of the hydrogen nuclei in the field is magnetized, which causes them to be distributed parallel to the direction of the main magnetic field. By applying a set of radio frequency pulses of characteristic frequencies to the main magnetic field, the low-energy hydrogen protons gain energy to enter the high-energy state. Consequently, the macroscopic magnetization is deflected and starts precession. Free induction decay (90-FID) and spin-echo pulse sequence (CPMG) are the two commonly used pulse sequences. After turning off the radio frequency pulse, the macroscopic longitudinal magnetization vector gradually returns to the equilibrium position before excitation under the effect of the main magnetic field. This process becomes longitudinal relaxation, also called spin–lattice relaxation. The macroscopic transverse magnetization vector gradually decays, a process called transverse relaxation or spin–spin relaxation. The time of lattice relaxation is *T*_1_ and the time of spin relaxation is *T*_2_ [45,46,47]. *T*_2_ can be calculated according to [46] as follows:(1)$$\frac{1}{T_{2}}=\frac{1}{T_{2,\mathrm{bulk}}}+\frac{1}{T_{2,\mathrm{surface}}}+\frac{1}{T_{2,\mathrm{diffusion}}}$$
where *T*_2_, bulk is the bulk relaxation time. Free water has a large bulk relaxation time, about seconds, and thus can be neglected. *T*_2_, surface is the surface relaxation time. *T*_20_ diffusion is the diffusion relaxation time, which can also be neglected when the magnetic field gradient is constant. Therefore, transverse relaxation time *T*_2_ can be approximated by *T*_2_, surface. The relationship between surface relaxation time with pore volume (*V*) and pore surface area (*S*) is calculated as follows:(2)$$\frac{1}{T_{2,\mathrm{surface}}}=\rho \frac{S}{V}$$
where *ρ* is the material surface relaxivity and takes the value of 12.5 nm/ms [48]. Based on the dimensional analysis, Equation (2) can be expressed as follows:(3)$$\frac{1}{T_{2,\mathrm{surface}}}=\rho k\frac{1}{r}$$
where *r* and *k* denote the pore radius and geometric factor of the pore, respectively. In this study, we assumed that the pore was columnar pore, and the value of *k* is 2. From Equation (3), it is evident that the transverse relaxation time of pore water is positively correlated with the pore size.

Testing of transverse relaxation time was conducted on a Macro MR12-150H-I (Suzhou, China), with a constant magnetic field of 0.3 ± 0.05 T, a magnet temperature of 32 ± 0.02 °C and a radio frequency coil diameter of 150 mm. The equipment working frequency was 12.638 MHz, and can test the size of samples that are less than 150 mm [43,49]. When the samples reached the test age 1 day before, they were removed from the molds, vacuum-saturated with water for 24 h, and then tested by NMR.

## 3. Results and Discussion

### 3.1. Compressive Strength

The compressive strength of manufactured sand concrete with different FA contents under the constant temperature stage of steam curing is shown in Figure 6. The strength of F0, F10 and F20 concrete could reach the demoulding strength requirement after 6 h of constant temperature curing, while F30 concrete needed 12 h. The strength of F0, F10 and F20 concrete could reach the prestressing tensioning strength requirement after 24 h of constant temperature curing, while F30 still could not reach the prestressing tensioning strength after 36 h of constant temperature curing. These results indicate that replacing part of the cement with FA can significantly extend the steam curing time required for concrete to reach the demoulding strength and prestressing tension strength. This time can be extended with increase in FA content. All FA concretes under the constant temperature stage of steam curing had lower compressive strengths than those of pure cement concrete under the same age. The FA admixture was increased from 10% to 30% and the strength of FA concrete was continuously reduced. The strength of FA concrete was negatively linearly correlated with the FA contents before 24 h of constant temperature curing, with the fitting coefficient *R*^2^ above 0.85; but after 24 h, *R*^2^ decreased to below 0.7. Notably, the greater the amount of FA replacement, the higher the degree of reduction in early strength under steam curing. The strength difference between FA concrete and pure cement concrete was reduced after 24 h of constant temperature curing. Subsequent XRD, SEM and *T*_2_ spectra analysis revealed that the reduced content of hydration products and increased total pore content were the main reasons for the low strength of FA concrete, and the increase in gel content by the pozzolanic reaction of FA caused the strength difference between FA concrete and pure cement concrete to gradually become smaller.

The compressive strength of manufactured sand concrete with different FA contents at 28 days and 90 days under both steam and standard curing is shown in Figure 7. From the results, it is evident that the compressive strengths of pure cement concrete at 28 and 90 days under steam curing was reduced by 6.8 and 7.2%, compared to those under standard curing. This indicates that the thermal effect of steam curing was unfavorable to the late strength, which was consistent with conclusions from previous studies [4,17]. These phenomena are mainly attributed to the uneven distribution of hydration products and an uncompact microstructure at high temperatures [13,15,16]. Concrete supplemented with FA had lower compressive strengths at 28 days compared to those of pure cement concrete (F0) under steam curing. However, at 90 days, the compressive strength of F10 was close to F0, that of F20 was higher than F0, while that of F30 was still much lower than that of F0. Notably, adding an appropriate amount of FA could improve the late strength of concrete under steam curing. We attribute this phenomenon to several possible reasons: (1) Crystalline FA filled the pores and reduced the content of large pores. (2) Pozzolanic reaction products were cross-bonded with the cement hydration products, which improved the strength of the gel system. (3) Steam curing advanced the occurrence time of the pozzolanic reaction of FA, increased the degree of pozzolanic reaction, produced more calcium silicate hydrate and calcium aluminate hydrate, filled a large number of pores and microcracks and improved the compactness of the microstructure. Excessive FA significantly reduced the late strength of the concrete. This was due to the large amount of FA, which reduced the true cement content, resulting in a reduction in hydration products. Additionally, since most FA were crystalline phases, this made FA less reactive than cement [20,50,51,52], resulting in a reduced gel content.

### 3.2. Microstructure

SEM images of manufactured sand concrete under the constant temperature stage of steam curing and the late stage are presented in Figure 8. To confirm that the pozzolanic reaction of FA occurred at the constant temperature stage, we observed surface morphology of FA particles at the constant temperature stage. Results showed that the external morphology of FA spherical particles of F20 concrete first changed from a smooth spherical to a speckled surface, before curing at a constant temperature for 12 h (Figure 8a,b). Cement‘s hydration products were precipitated on the surface of FA, indicating that FA exerted a seeding effect. After 24 h of constant temperature curing, the surface of FA particles transformed into a fibrous mesh containing a large number of pores (Figure 8c,d), indicating that the pozzolanic reaction of FA had occurred. Pozzolanic reactions of FA under steam curing appeared at about 24 h of constant temperature curing, while the pozzolanic reaction appeared at about 7 days under 20 °C curing [26,53,54]. Our results further demonstrated that steam curing could significantly advance the pozzolanic reaction of FA.

Occurrence of the pozzolanic reaction of FA under the constant temperature stage during steam curing can be attributed to three reasons. Firstly, accelerated hydration of cement under high-temperature curing and early production of large amounts of calcium hydroxide provides the preconditions for pozzolanic reactions. Secondly, a large amount of calcium hydroxide produced over a short period of time increased alkalinity of the pore solution. The high pH facilitated the dissolution of active SiO_2_ and Al_2_O_3_ in the glassy FA, thereby accelerating pozzolanic reactions [51]. Lastly, the rapidly produced hydration and pozzolanic reaction products were distributed in a fibrous network on the surface of FA (Figure 8c,d), subsequently forming a large number of pores, which facilitated entry of Ca^2+^ and OH^−^ into the interior of FA at a later stage and promoted the pozzolanic reaction.

Under steam curing, pure cement concrete exhibited coarse and numerous microscopic pores, which were accompanied by uneven distribution of hydration products and the presence of microcracks (Figure 8e,f). These may explain the loss of late strength in steam-cured concrete [6,11,15]. In FA concrete, the inert FA mainly filled the pores and reduced the large pores content. The gel generated by the pozzolanic reaction of glassy FA not only filled the pores, but also cross-bonded with the C-S-H (Figure 8d), C-(A)-S-H (Figure 8h), needle-like AFt (Figure 8g) and plate-like CH (Figure 8d). Overall, these phenomena not only increased the strength of the gel system, but also improved microstructural compactness. That decreased the negative impact of steam curing on the microstructure, which in turn reduced the thermal damage to the strength.

### 3.3. XRD Analyses

Next, we used XRD to explore the effect of FA on the hydration of manufactured sand concrete during the constant temperature stage of steam curing. XRD patterns of F20 concrete cured at constant temperature for 6, 12, 24 and 36 h are shown in Figure 9. At 6 h of constant temperature curing, we observed both unhydrated C_3_S (PDF #49-0442) and C_2_S (PDF #49-1673) in all four groups of concrete specimens. The diffraction peak of ettringite (PDF #41-1451) can also be observed, which was a hydration product of tricalcium aluminate (C_3_A). Gismondine (PDF #39-1373) can be observed, but the intensity of its diffraction peak was low, which was a product of cement hydration [55]. Moreover, CH (PDF #04-0733) diffraction peaks were present. The intensity of the characteristic peak of CH in F30 was the lowest, and the intensity of the characteristic peak of C_3_S in F0 was the lowest. According to the principle of the hydration reaction of Portland cement, cement hydration degree can be expressed by the content of CH [50,53]. The actual cement content in FA concrete was reduced, resulting in less early CH production and therefore lower early compressive strength.

The Intensity of the diffraction peaks of C_3_S and C_2_S significantly decreased in all four groups of specimens, while that of the diffraction peaks of CH increased after 12 h of constant temperature curing. This indicated the intense occurrence of the hydration reaction of the cement. The intensity of the CH diffraction peak in F30 was not significantly reduced, indicating that the pozzolanic reaction of FA did not appear. The diffraction peaks of C_3_S and C_2_S were still observed at a constant temperature of 24 h, but the intensity of their diffraction peaks was significantly lower. The intensity of the diffraction peak of CH continued to increase, with the largest increase observed in F0. The low increase in CH diffraction peak intensity in FA concrete may be attributed to the following reasons: Firstly, addition of FA resulted in lower cement content in concrete and reduced the amount of CH produced by hydration. Secondly, part of the CH produced by cement hydration started to be consumed by the pozzolanic reaction of FA, as evidenced by SEM images. We found no obvious change in the intensity of diffraction peaks of CH, C_2_S and C_3_S across all four groups of samples after 36 h under constant temperature curing. This indicated that the hydration reaction became significantly slower.

Profiles of XRD patterns during later stages of steam-cured manufactured sand concrete are illustrated in Figure 10. From the results, it was observed that the intensity of the diffraction peak of CH decreased with an increase in FA content. Notably, higher FA content resulted in more CH consumption, a higher degree of FA reaction, which is consistent with the regularity under ambient temperature curing [51]. The diffraction peaks of C_2_S and C_3_S were still observed at 28 days and 90 days, but the intensity of the diffraction peak at 90 days was lower. This indicated that C_2_S and C_3_S were still slowly hydrated, and the CH produced by late hydration also provided the conditions for the persistence of the pozzolanic reaction.

### 3.4. T_2_ spectra

The area integral of the *T*_2_ spectral distribution curve reflected the pore volume of the concrete, and the relaxation time represented the pore size [45,48]. We divided the pore size to gel pores (d < 10 nm), capillary pores (10 nm–1000 nm) and large pores (d > 1000 nm) [49]. Distribution curves of *T*_2_ spectra of steam-cured manufactured sand concrete at 28 and 90 days with different FA contents are shown in Figure 11. Based on the results, it was observed that the first peak in the *T*_2_ spectrum in FA concrete had significantly higher relaxation-signal intensity and spectral area than those of pure cement concrete. The peak signals of the second and third peaks shifted to the left. This indicated that the volume of both gel and capillary pores increased, while the pore size of large pores decreased in FA concrete compared to pure cement concrete. Based on SEM images, we attributed this phenomenon to the pozzolanic reaction and filling effect of FA. Increase in age caused a decrease in the peak signal intensity of *T*_2_ spectra in all four groups of specimens. Moreover, the peak signals of the second and third peaks moved to the left, with a large left shift in the FA concrete. These results indicated that both the subsequent continuous hydration and pozzolanic reaction could reduce the pore content and refine the pore size. FA concrete had a dual effect of hydration and pozzolanic reaction, which subsequently generated a significant variation in the peak signal intensity and location.

The area integral of the *T*_2_ spectrum distribution curves of manufactured sand concrete with different FA contents under steam curing are shown in Figure 12. Summarily, FA concrete had higher total pore volumes under steam curing than pure cement concrete, which was consistent with conclusions from previous studies under 20 °C curing [27,56]. This phenomenon was attributed to the increase in the real water-to-binder ratio in FA concrete. Total pore volume first decreased, then increased when the FA content in concrete increased from 10 to 30%, indicating that there was an optimum value of FA addition. Combined with SEM and XRD images, it was evident that FA‘s filling effect and pozzolanic effect caused a decrease in large pore content and an increase in microstructural compactness, thereby favorably affecting the strength. Increase in age caused a marked reduction in total pore volume of the FA concrete relative to that of pure cement concrete, indicating that the pozzolanic reaction during later stages significantly affected the reduction in the pore volume.

The ratio of the area of the first, second and third peaks to the total integrated area in the *T*_2_ spectra of manufactured sand concrete with different FA contents is illustrated in Figure 13. Summarily, the largest percentage of the first peak area indicated that most of the pores in the steam-cured manufactured sand concrete were both gel and capillary pores. Notably, an increase in FA content generated a corresponding increase in the proportion of the first peak area, and a decrease in the proportion of the second and third peak areas. These results demonstrated that the volume proportion of gel pores and capillary pores in manufactured sand concrete increased while the proportion of large pores decreased.

## 4. Conclusions

In summary, our findings indicate that the addition of appropriate amounts of FA can alleviate the negative effects of steam curing on the long-term strength of manufactured sand concrete. This is expected to facilitate production of precast concrete beams in cold regions. The following main conclusions are drawn from this study:(1)The steam curing time required to reach the demoulding and prestress tensioning strengths of F30 was 100% and 50% longer than that of F0, respectively. This phenomenon was caused by the dilution effect of FA.(2)The intensity of the diffraction peak of CH in FA concrete decreased after 24 h of constant temperature under steam curing. FA particles were dissolved and the smooth spherical surface changed to a fibrous network, which proved that the pozzolanic reaction of FA occurred, indicating that steaming could remarkably advance the pozzolanic reaction time of FA.(3)Appropriate amounts of FA need to be added in the steam-cured manufactured sand concrete. The reason was that the filling effect and pozzolanic reaction of FA would result in lower large pore content, increased microstructural compactness and higher strength of the gel system, which in turn reduced the adverse effect of steam curing on the microstructure, and therefore had a beneficial effect on the long-term strength. The amount of FA admixture in steam-cured manufactured sand concrete was suitable to be controlled at 20%.

Although it was proved that the pozzolanic reaction of FA could happen at 24 h of steam curing, the following aspects still need further study: (1) Reaction kinetics of FA concrete under steam curing [19]. (2) Effect of FA on other properties of steam-cured manufactured sand concrete, such as chloride ion impermeability and frost resistance.

## Figures and Tables

**Figure 1 materials-16-01326-f001:**
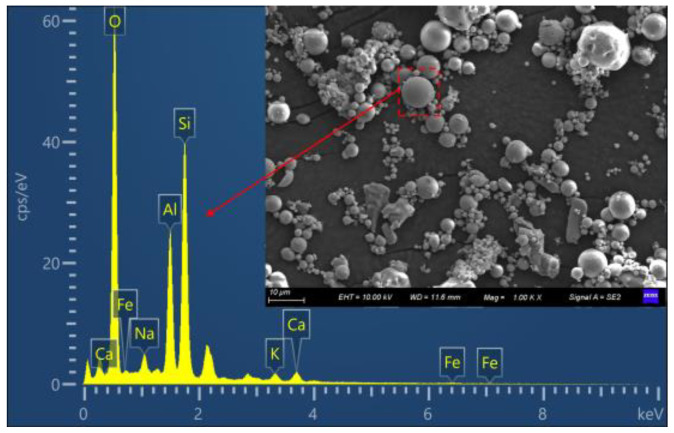
SEM and EDS of FA.

**Figure 2 materials-16-01326-f002:**
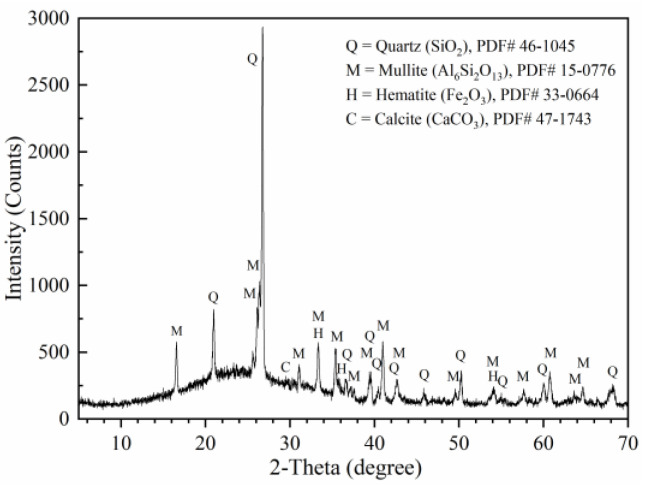
XRD of FA.

**Figure 3 materials-16-01326-f003:**
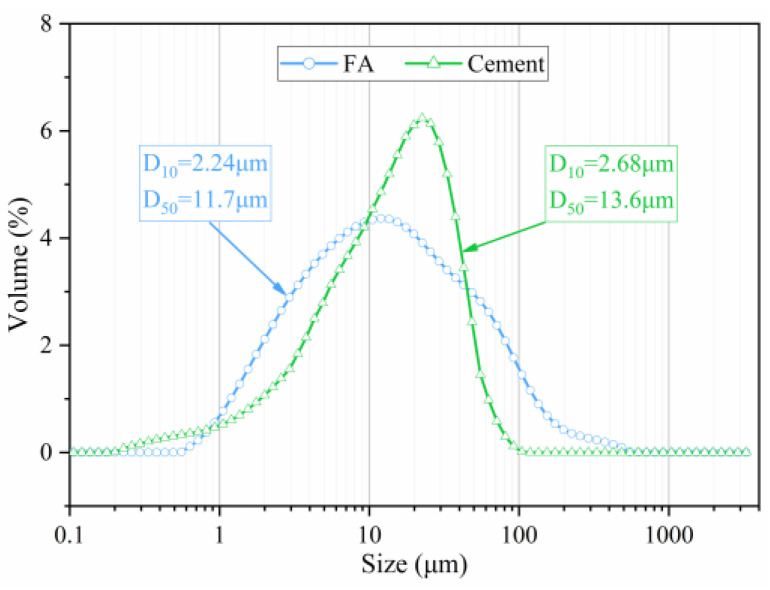
Particle size distributions of OPC and FA.

**Figure 4 materials-16-01326-f004:**
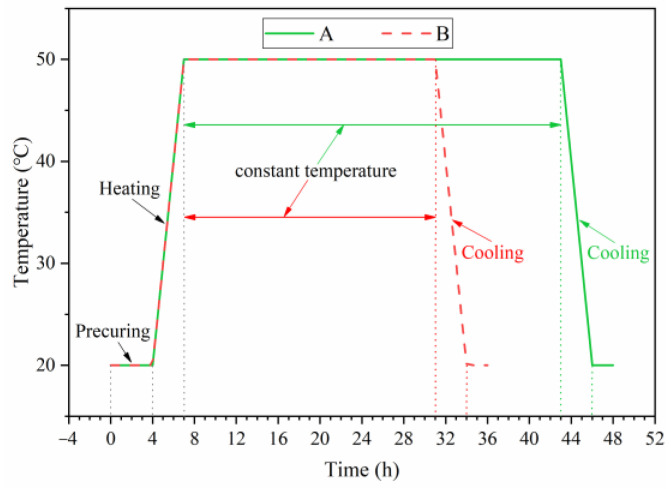
Steam curing regime.

**Figure 5 materials-16-01326-f005:**
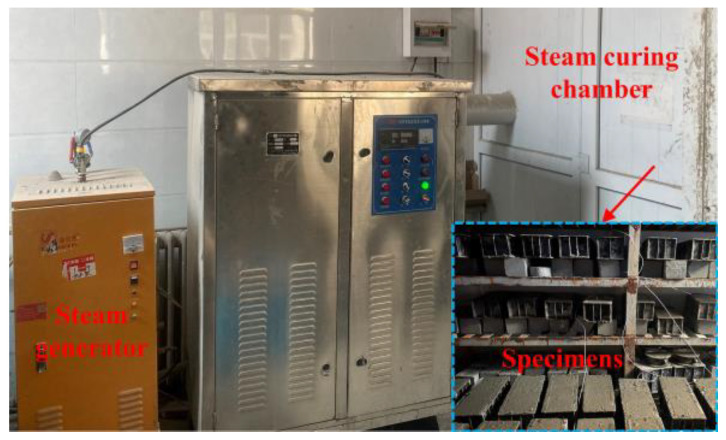
Steam curing equipment.

**Figure 6 materials-16-01326-f006:**
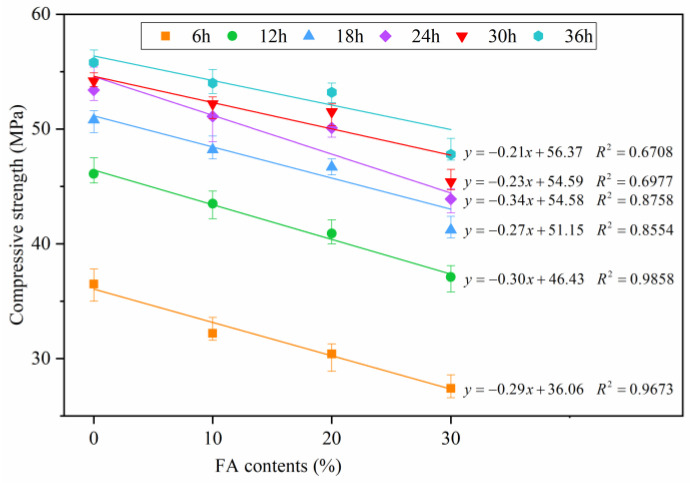
Relationship between FA concrete compressive strength and FA content.

**Figure 7 materials-16-01326-f007:**
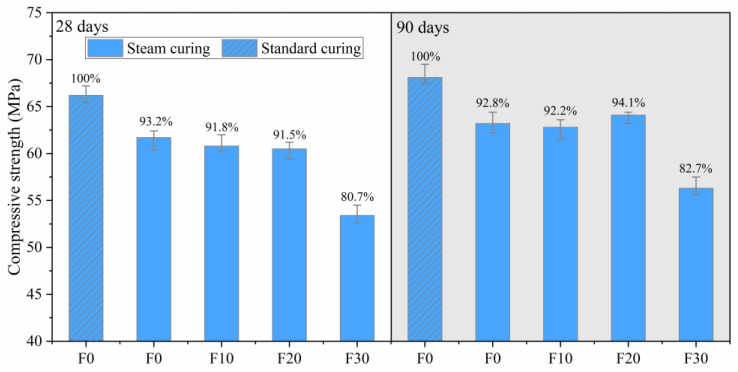
Compressive strength of concrete with different FA contents at 28 days and 90 days.

**Figure 8 materials-16-01326-f008:**
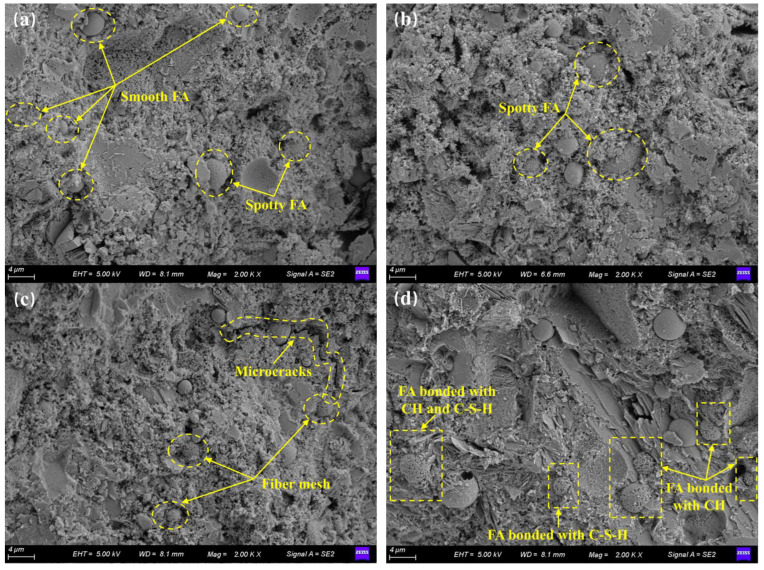
SEM image of concrete. Images (**a**–**d**) are F20 maintained under steam curing for 6 h, 12 h, 24 h and 36 h, respectively. Images (**e**,**f**) are 24 h and 28 days after steam curing of F0. Images (**g**,**h**) are 28 days and 90 days after steam curing of F20.

**Figure 9 materials-16-01326-f009:**
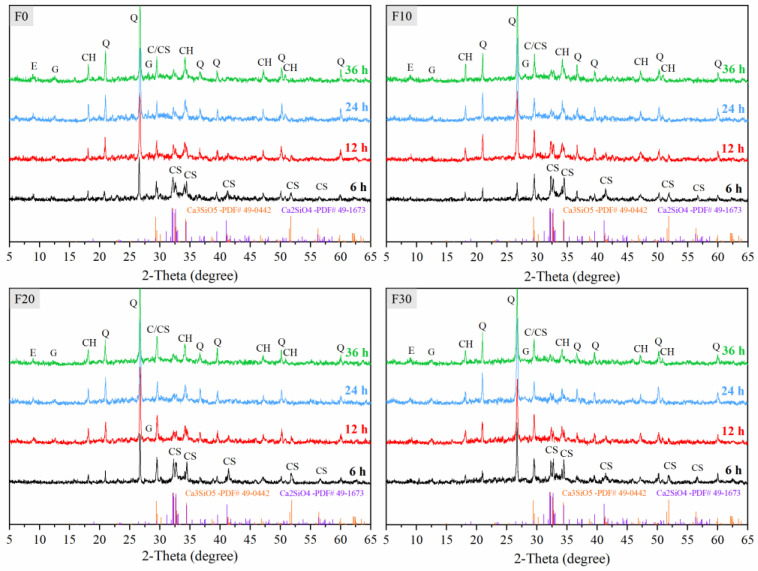
XRD at constant temperature under steam curing. E–Ettringite; CH–Portlandite; Q–Quartz; C–Calcite; CS–C_3_S/C_2_S; G–Gismondine.

**Figure 10 materials-16-01326-f010:**
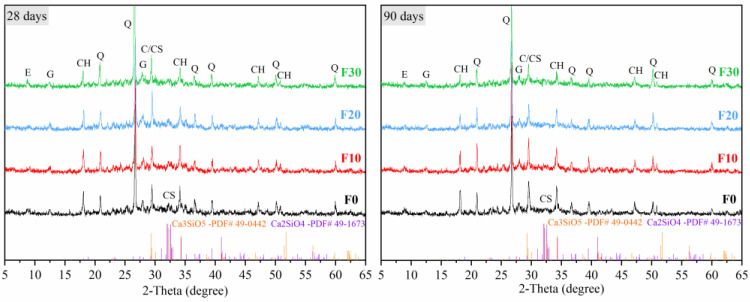
XRD after steam curing. E–Ettringite; CH–Portlandite; Q–Quartz; C–Calcite; CS–C_3_S/C_2_S; G–Gismondine.

**Figure 11 materials-16-01326-f011:**
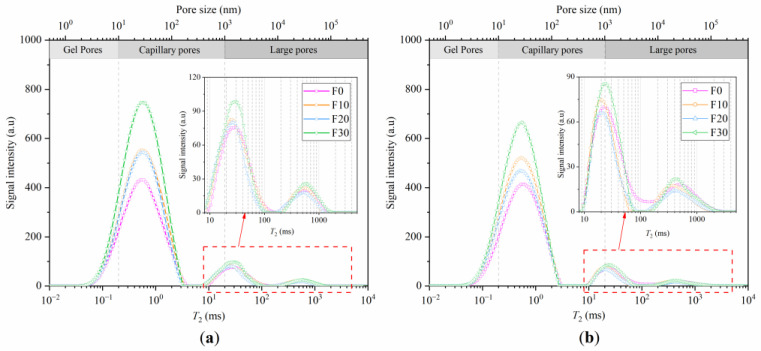
*T*_2_ spectrum distribution curves of concrete with different FA contents. (**a**) At 28 days, (**b**) At 90 days.

**Figure 12 materials-16-01326-f012:**
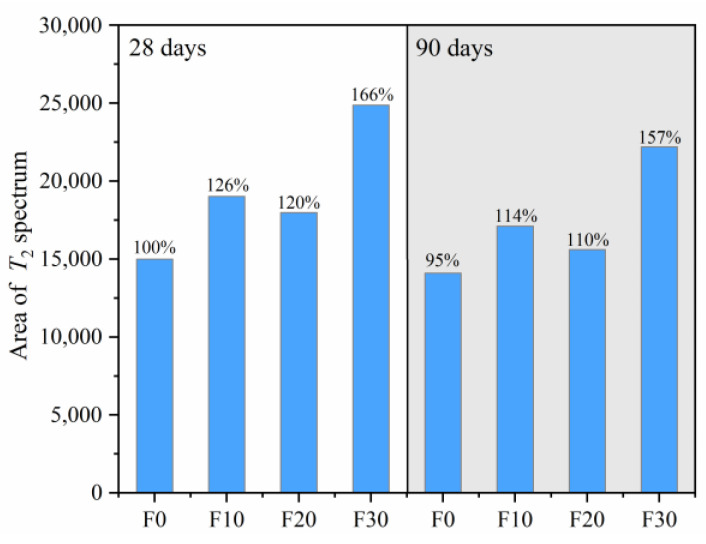
Integral area of the *T*_2_ spectrum.

**Figure 13 materials-16-01326-f013:**
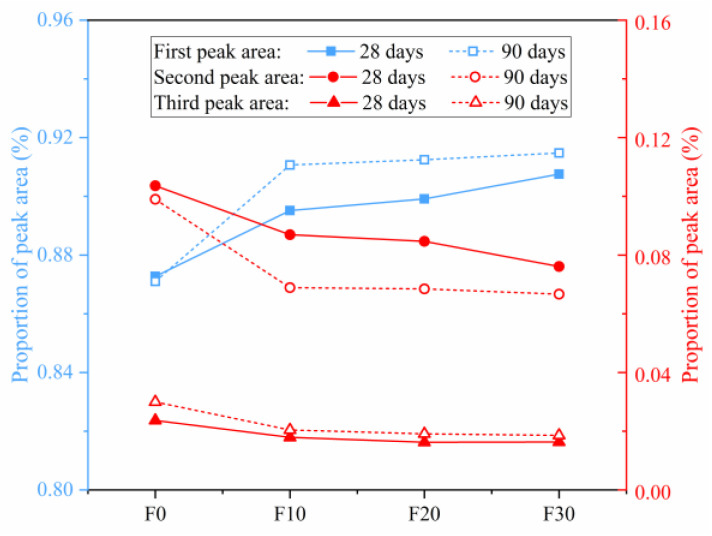
The proportion of each peak.

**Table 1 materials-16-01326-t001:** Chemical compositions of Portland cement and FA.

Chemical Compositions (% by Mass)	SiO_2_	CaO	Al_2_O_3_	Fe_x_O_y_	K_2_O	MgO	Na_2_O	SO_3_	TiO_2_	Loss of Ignition
PC	22.06	61.56	4.45	3.46	0.68	2.54	0.31	2.67	0.37	1.88
FA	54.92	5.91	30.76	3.07	1.14	0.88	0.51	0.08	-	2.56

**Table 2 materials-16-01326-t002:** Mix proportion of concrete with different FA contents.

No.	Mix Proportion (kg/m^3^)					
	Water	Cement	Fly Ash	Crushed Stone	Manufactured Sand	Superplasticizer
F0	157	490	0	1148	704	5.3
F10	157	441	49	1148	704	5.3
F20	157	392	98	1148	704	5.3
F30	157	343	147	1148	704	5.3

## Data Availability

Not applicable.

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
