# Peer review of "Investigating the Impact of Fly Ash on the Strength and Micro-Structure of Concrete during Steam Curing and Subsequent Stages"

_materials, 2023, doi:10.3390/ma16041326_

Round 1
Reviewer 1 Report
The manuscript entitled "Effect of fly ash on the strength and microstructure of manufactured sand concrete at the steam curing stage and later" presents an interesting study conducted on the characterization of concrete with FA addition cured at 50 °C. However, the description of some experiments wasn’t clearly presented, and some other issues must be addressed. The paper needs major revisions before it is processed further. Some comments follow:
Title: The title can be improved, as it is currently too long; maybe the authors can remove the formulation "manufactured sand."
Materials and methods section
Chemical compositions of Portland cement and FA: Two types of iron oxides have been detected in these types of materials; therefore, please replace Fe2O3 with FexOy in the XRF analysis or provide the scientific proof to support your results (XRD that show the presence or absence of magnetite (Fe2O3) or hematite (Fe3O4)).
XRD of FA: It is surprising to see that FA contains only these two phases. Could authors provide the ID of the phases identified by the XRD software so the reviewers may check the correspondence between the peaks and the attributes provided by the authors?
Concrete mixing and specimen preparation—What was the rationale for choosing these parameters? such as the amount of FA, the concentration of superplasticizer, and so on? Are these the most relevant mixtures considering the state-of-art in this field. The authors should provide a reason for this choice and cite relevant studies that support the affirmations.
The authors state, "The fresh concrete was loaded into a cube mold, measuring 100mm 100mm 100mm, for compressive strength and NMR testing." This is also very surprising since the diameter of the Macro MR12-150H-I NMR equipment (where the samples are placed) is around 15 mm. Please remove the ambiguity.
The NMR experiments have to be conducted on specific samples; otherwise, the results aren’t relevant.
XRD analysis: There are multiple peaks present in the XRD pattern that haven’t been considered. Why do the authors consider some peaks instead of others (there are some clear peaks around 12, 37, 57, 62, etc.)? Please improve the description of the XRD spectra.
Author Response
Dear Reviewer:
We appreciate all of the valuable comments from the reviewer of our work. We have revised our manuscript according to the reviewer’s comments questions, and suggestions. We highly appreciate the reviewer for their insightful comments and criticism, which have helped us improve both the content and the presentation of our work. We made sure that the reviewer's comments has been addressed carefully and the paper was revised accordingly.
(1) Title: The title can be improved, as it is currently too long; maybe the authors can remove the formulation "manufactured sand."
Response: According to the suggestions of two reviewers, the title is revised as follows: Investigating the impact of fly ash on the strength and microstructure of concrete during steam curing and subsequent stages
(2) Materials and methods section
Chemical compositions of Portland cement and FA: Two types of iron oxides have been detected in these types of materials; therefore, please replace Fe2O3 with FexOy in the XRF analysis or provide the scientific proof to support your results (XRD that show the presence or absence of magnetite (Fe2O3) or hematite (Fe3O4)).
Response: Fe2O3 is replaced by FexOy in the XRF analysis. Meanwhile Table 1 has been revised.
(3) XRD of FA: It is surprising to see that FA contains only these two phases. Could authors provide the ID of the phases identified by the XRD software so the reviewers may check the correspondence between the peaks and the attributes provided by the authors?
Response: The authors re-analyzed the XRD patterns of FA, which contains four phases, namely Quartz (SiO2 PDF# 46-1045), Mullite (Al6Si2O13 PDF# 15-0776), Hematite (Fe2O3 PDF# 33-0664), and Calcite (CaCO3 PDF# 47-1743). The ID of the phases identified by the XRD software is provided. Figure 2 in the article has also been revised.
(4) Concrete mixing and specimen preparation—What was the rationale for choosing these parameters? such as the amount of FA, the concentration of superplasticizer, and so on? Are these the most relevant mixtures considering the state-of-art in this field. The authors should provide a reason for this choice and cite relevant studies that support the affirmations.
Response: Thanks to the comments of the reviewers, the authors have given the basis for the choice of concrete mix proportions, FA contents, and superplasticizer concentrations in the manuscript. The text was revised as following: Normally, a maximum of 35% by mass of FA can be used to replace cement in concrete [19,31], so the admixture of FA by mass in the concrete mixes of this study was 10%, 20% and 30%. Details of the mixing proportions of the concrete with different FA contents are shown in Table 2. This mix proportion (F0) was the common ratio for precast beams of high-speed railway in China [41]. The polycarboxylate superplasticizer with solids content of 23.6% and water reduction rate of 27% was used to improve the workability of the manufactured sand concrete and achieve the expected slump requirements [41].
(5) The authors state, "The fresh concrete was loaded into a cube mold, measuring 100mm×100mm×100mm, for compressive strength and NMR testing." This is also very surprising since the diameter of the Macro MR12-150H-I NMR equipment (where the samples are placed) is around 15 mm. Please remove the ambiguity.
Response: We use the newest NMR equipment for pore structure testing. The Macro MR12-150H-I MRI equipment can test samples smaller than 150 mm in size (cylinder or rectangular samples), while citing existing literature ([43] and [49]) to support it. The text was revised as following: The equipment working frequency was 12.638 MHz, can test the size of less than 150mm samples [43,49].
(6) The NMR experiments have to be conducted on specific samples; otherwise, the results aren’t relevant.
Response: NMR experiments have been conducted on specific samples, and we have added a description of this in section 2.4.4. The text was revised as following: When the samples reached the test age 1 day before, they were removed from the molds, vacuum saturated with water for 24h, and then tested by NMR.
(7) XRD analysis: There are multiple peaks present in the XRD pattern that haven’t been considered. Why do the authors consider some peaks instead of others (there are some clear peaks around 12, 37, 57, 62, etc.)? Please improve the description of the XRD spectra.
Response: According to the reviewer's comments and the theory of hydration reaction of fly ash-cement system (detailed description in the Introduction), we have added some phase analysis. The XRD analysis in this paper was to further confirm that the pozzolanic reaction of FA occurs at the constant temperature stage of steam curing and to investigate the degree of cement hydration at this stage, therefore we mainly discussed the changes of CH, C3S and C2S diffraction peaks. The characteristic peaks of gismondine (PDF #39-1373) correspond to 2θ of 12.114 and 27.982. The characteristic peaks of ettringite (PDF #41-1451) correspond to 2θ of 9.091 and 15.784. The peaks around 37, 57, 62 were the diffraction peaks of C3S and C2S, which we have annotated at the bottom of both Figure 9 and Figure 10. The text was revised as following: The diffraction peak of ettringite (PDF #41-1451) can also be observed, which was a hydration product of tricalcium aluminate (C3A). Gismondine (PDF #39-1373) can be observed, but the intensity of its diffraction peak was low, which was a product of cement hydration [55].
Please see the attachment for the revised manuscript
Reviewer 2 Report
Check the attached file

Author Response
Dear Reviewer:
We appreciate all of the valuable comments from the reviewer of our work. We have revised our manuscript according to the reviewer’s comments questions, and suggestions. We highly appreciate the reviewer for their insightful comments and criticism, which have helped us improve both the content and the presentation of our work. We made sure that the reviewer's comments has been addressed carefully and the paper was revised accordingly.
(1) The title could be written as follows:
"Investigating the Impact of Fly Ash on the Strength and Microstructure of Manufactured Sand Concrete during Steam Curing and Subsequent Stages"
Response: According to the suggestions of two reviewers, the title is revised as follows: Investigating the impact of fly ash on the strength and microstructure of concrete during steam curing and subsequent stages
(2) It has been observed that the majority of the references cited in the introduction are outdated. In order to enhance the credibility and relevance of the work, it is recommended to incorporate more recent references. One potential resource that may be worth considering is the following:
Khatib, J., Jahami, A., El Kordi, A., Sonebi, M., Malek, Z., Elchamaa, R. and Dakkour, S. (2021). Effect of municipal solid waste incineration bottom ash (MSWI-BA) on the structural performance of reinforced concrete (RC) beams. Journal of Engineering, Design and Technology, https://doi.org/10.1108/JEDT-01-2021-0068.
Response: Thanks to the reviewer for recommending references to us. We have replaced a few references in the original manuscript with the latest references.
(3) It is recommended to establish a correlation between the compressive strength and the percentage of fly ash by creating a fitting graph.
Response: Based on the reviewer's comments we revised Figure 5 in the original manuscript. We established a fitted relationship between FA concrete compressive strength and FA percentage. The following text was added: The FA admixture was increased from 10% to 30% and the strength of FA concrete was continuously reduced. The strength of FA concrete was negative linearly correlated with the FA contents before 24 h of constant temperature curing, with the fitting coefficient R2 above 0.85, but after 24 h, R2 decreased to below 0.7.
(4) In the conclusion section, present the findings in a quantitative manner, expressed as percentages. Additionally, it is suggested to provide recommendations for further research to expand upon the current study.
Response: According to the reviewer's comments, the first conclusion was presented in a quantitative manner. The text was revised as following: The steam curing time required to reach the demoulding and prestress tensioning strengths of F30 was 100% and 50% longer than that of F0, respectively. This phenomenon was caused by the dilution effect of FA.
The authors also give suggestions for further research, as follows: Although it was proved that the pozzolanic reaction of FA could happen at 24 h of steam curing, the following aspects still need further study: (1) Reaction kinetics of FA concrete under steam curing [19]. (2) Effect of FA on other properties of steam-cured manufactured sand concrete, such as chloride ion impermeability and frost resistance.
Please see the attachment for the revised manuscript.
Reviewer 3 Report
This research paper investigates the effect of eco-friendly binder on the microstructure and strength development of steam curing concrete. The article is well-written and deserved publication in materials journal after a minor revision.
· The abstract contain too much information about the environmental benefits of fly ash. In general the abstract is too long. Must reduce its length. It should be crispy and concise to increase the interest in readers.
· In abstract, mention about the percentage of fly ash.
· L38, no need for three references for a single piece of information
· L43, same as the previous comment for this information and throughout the paper.
· In the introduction. The paragraphs are quite long. Good research articles focus concerning to your on study on the main points avoiding too many details.
· Lines 56-80 the paragraph must be encapsulated.
· Also mention the fly ash type (Class F/C)
· The problem statement is weak.
· Line 101: when you mention many research exist. You need to cite them as well. Or point to the above paragraphs (by writing as mentioned above).
· SEM image must be enlarged.
· What type of coal is used as a fuel to generate this FA. And mention the class of fly ash.
· Include the picture of apparatus used for steam curing.
· Results and discussion section is well written. T2 spectra results need some validation.
Author Response
Dear Reviewer:
We appreciate all of the valuable comments from the reviewer of our work. We have revised our manuscript according to the reviewer’s comments questions, and suggestions. We highly appreciate the reviewer for their insightful comments and criticism, which have helped us improve both the content and the presentation of our work. We made sure that the reviewer's comments has been addressed carefully and the paper was revised accordingly.
(1) The abstract contain too much information about the environmental benefits of fly ash. In general, the abstract is too long. Must reduce its length. It should be crispy and concise to increase the interest in readers.
Response: The information about the environmental benefits of fly ash was removed. We also revised the abstract to make it concise.
(2) In abstract, mention about the percentage of fly ash.
Response: In abstract, the percentage of fly ash was mentioned. The text was revised as following:
In this paper, we investigated the effect of FA (10%~30%) on the compressive strength and microstructure of manufactured sand concrete at steam curing stage and later.
The addition of 30% FA to manufactured sand concrete causes a significant reduction in early strength under steam curing, which is not beneficial to the formwork removal and tensioning of precast members.
Therefore, the addition of 20% FA to the manufactured sand concrete can improve the long-term strength, which is beneficial to production of precast beams in cold regions.
(3) L38, no need for three references for a single piece of information
Response: We have simplified the citation of this information by adopting two references.
(4) L43, same as the previous comment for this information and throughout the paper.
Response: We have unified this information throughout the paper, i.e., demoulding strength.
(5) In the introduction. The paragraphs are quite long. Good research articles focus concerning to your on study on the main points avoiding too many details.
Response: Based on the reviewers' comments, we have revised the introduction, especially the second paragraph. Some details of the description have been removed. Detailed modification information was shown in the introduction.
(6) Lines 56-80 the paragraph must be encapsulated.
Response: We have encapsulated lines 56-80 of the original manuscript.
(7) Also mention the fly ash type (Class F/C)
Response: In the third paragraph of the introduction, we have added the type of FA (Class F/C) to the description of the relevant content.
(8) The problem statement is weak.
Response: We have restated the problem. The text was revised as following:
As mentioned above, although many researches have been conducted in this field, several limitations still exist. For example, controlling constant temperature duration in the current steam curing regime does not effectively meet the tensioning requirements of precast beams, especially for the production of precast beams in cold regions. Studies on FA concrete have mainly focused on normal temperature curing, and there is a lack of research on the strength and microstructure of FA concrete under steam curing, especially the steam curing process, because it is a critical period for the formation of concrete properties and microstructure. It is certain that high temperature curing can have harmful effects on the long-term strength and microstructure of concrete, but only a few studies have explored how to reduce the thermal damage effect.
(9) Line 101: when you mention many research exist. You need to cite them as well. Or point to the above paragraphs (by writing as mentioned above).
Response: The information in line 101 of the original manuscript has been revised based on the suggestions of the reviewer. The text was revised as following: As mentioned above, although many researches have been conducted in this field, several limitations still exist.
(10) SEM image must be enlarged.
Response: The SEM image has been enlarged.
(11) What type of coal is used as a fuel to generate this FA. And mention the class of fly ash.
Response: We have added this information in section 2.1. The following text was added: Class F type of FA was generated by burning bituminous coal, which was obtained from Xigu Power Plant, China.
(12) Include the picture of apparatus used for steam curing.
Response: We have added picture of the steam curing equipment in section 2.3.
(13) Results and discussion section is well written. T2 spectra results need some validation.
Response: We validated the T2 spectral results by citing references. The text was revised as following: The area integral of the T2 spectral distribution curve reflected the pore volume of the concrete, and the relaxation time represented the pore size [45,48]. We divided the pore size to gel pores (d < 10 nm), capillary pores (10 nm-1000 nm) and large pores (d > 1000 nm) [49].
Please see the attachment for the revised manuscript.
Round 2
Reviewer 1 Report
Dear authors,
You have done a great job revising your paper. I don't have any further recommendations to improve your article.
Best regards,
Reviewer 2 Report
No more comments